# The Relationship between Short- and Long-Term Memory Is Preserved across the Age Range

**DOI:** 10.3390/brainsci13010106

**Published:** 2023-01-05

**Authors:** Giedrė Čepukaitytė, Jude L. Thom, Melvin Kallmayer, Anna C. Nobre, Nahid Zokaei

**Affiliations:** 1Oxford Centre for Human Brain Activity, Wellcome Centre for Integrative Neuroimaging, Department of Psychiatry, University of Oxford, Oxford OX3 7JX, UK; 2Department of Experimental Psychology, University of Oxford, Oxford OX2 6GG, UK; 3Department of Psychology, Goethe University, 60323 Frankfurt, Germany

**Keywords:** short-term memory, long-term memory, spatial-contextual memory, ageing

## Abstract

Both short- and long-term memories decline with healthy ageing. The aims of the current study were twofold: firstly, to build on previous studies and investigate the presence of a relationship between short- and long-term memories and, secondly, to examine cross-sectionally whether there are changes in this relationship with age. In two experiments, participants across the age range were tested on contextual-spatial memories after short and long memory durations. Experimental control in stimulus materials and task demands enabled the analogous encoding and probing for both memory durations, allowing us to examine the relationship between the two memory systems. Across two experiments, in line with previous studies, we found both short-term memory and long-term memory declined from early to late adulthood. Additionally, there was a significant relationship between short- and long-term memory performance, which, interestingly, persisted throughout the age range. Our findings suggest a significant degree of common vulnerability to healthy ageing for short- and long-term memories sharing the same spatial-contextual associations. Furthermore, our tasks provide a sensitive and promising framework for assessing and comparing memory function at different timescales in disorders with memory deficits at their core.

## 1. Introduction

Memories at short and long timescales are highly interconnected. Information extracted from the environment and held over the short term ultimately forms our long-term memories [1,2]; in turn, our past experiences bias our short-term memories [3,4,5]. However, few studies have specifically explored the co-variations between short-term memory (STM) and long-term memory (LTM) for the same material [6,7,8,9,10,11,12,13,14,15]. In the case of spatial-contextual associations, for example, both memory functions decline with advancing age [16,17,18,19,20,21,22], and a greater degree of memory preservation in both cases indicates healthy ageing [23,24,25]. This broad pattern suggests a shared vulnerability of short- and long-term memories to age-related processes. 

Spatial-contextual short- and long-term memories may share some underlying neural mechanisms, resulting in the observed vulnerability of both memory functions to ageing. For example, the essential role of the medial temporal lobe (MTL) in episodic LTM has been evident at least since the early lesion studies, which showed that individuals with damage to this part of the brain, including the hippocampus, could not form new long-term memories that can be consciously accessed [26,27,28]. Recent studies have shown that the MTL is also crucial for retaining features, specifically object-location associations, bound together in STM [29,30,31,32]. Deficits are prominent either when two or more items have to be remembered or when maintaining the exact bindings between objects, and their location is necessary for successful performance [29,30,31,32]. As a result, it has been proposed that the MTL is involved in the relational binding of representations, irrespective of the memory duration [31,33], although see [34,35].

To test whether ageing-related changes influence a shared set of processes critical to both memory durations, it is useful to test whether the pattern of co-variation remains consistent over a broad age range. Although similar patterns of deficits have been noted, studies have not compared performance for STM and LTM directly at multiple ages while also controlling for extraneous factors that could affect performance. To chart the relation between STM and LTM, it is essential to vary only the lifetime of the memory, while keeping the contents of the memoranda and task demands the same. Due to different stimulus materials, task requirements, and output measures across studies, the extent of the overlap between STM and LTM, and how they covary with healthy ageing, remains unclear. 

Studies attempting to probe memories derived from common encoding tasks are few and often test short- and long-term memories using different methodologies [36,37]. For example, in one study, performance on a delayed match-to-sample STM task in young, middle-aged, and older adults was compared to two-alternative forced-choice surprise LTM recognition [37]. Alternatively, another study that assessed young and older adults contrasted memory for item identity in a two-back STM task with LTM recognition of incidentally encoded object-quadrant associations [36]. Not surprisingly, results across studies have led to a mixed picture of the co-variation between STM and LTM among age groups, with one study finding no differences [37] and the other finding a reduced correlation between STM and LTM in older adults [36].

Other studies have utilised similar but not identical tasks to encode analogous STM and LTM associations. For example, a study by [38] tested associative memories for object-colour bindings at both short and long durations. Participants performed two separate continuous-report tasks. In the STM task, they were first shown a display with three coloured objects followed by a memory delay of one second. Following the delay, participants had to adjust the colour of one of the objects, probed by location, to match that of the original object in the memory array. In contrast, in the LTM task, participants viewed five displays of three coloured objects one at a time, separated by an interval of one second. The colour-report task then followed a 30-s delay. Participants had to recall and adjust the colours of all 15 objects they had observed in the preceding displays. They found that LTM precision declined disproportionately with age compared to STM precision [38]. Although these tasks probed analogous associations in both timescales, and their continuous-report nature added significant sensitivity and granularity to the measurements, the relationship between these two memory functions may have been understated. Despite similarities between the two tasks, important details may have caused differences in encoding strategy and response variability. STM was probed for one item out of three held in memory, while all fifteen items retained in long-term memory were probed sequentially. This sequential retrieval of object-colour associations could further have added more variability to the LTM as compared to the STM performance, possibly due to greater interference from previously recalled stimuli [39,40,41].

One recent study has managed to overcome these issues by testing short- and long-term memories for object identities in young and older adults [42]. The task involved showing participants two, four or six icons and then assessing their memories for these items either at the end of each trial, two seconds after the presentation of the stimuli (STM), or in a subsequent task (LTM), separated by a one-minute distractor task. In both cases, participants’ memories were assessed by showing them an object, which was either novel or an item from the memory array in 50% of trials and asking them to indicate their response by rating their confidence on a six-item scale. The authors found that although both short- and long-term memories were impaired in older adults compared to their young counterparts, the degree of impairment from STM to LTM was equivalent in both groups [42]. They interpreted their findings as supporting a shared encoding bottleneck between the two memory systems [42]. However, in this study, memories were not reproduced explicitly, allowing responses based on familiarity. A task enabling a more granular assessment of STM and LTM may reveal a different pattern of results.

The aims of the current study were to explore the relationship between short- and long-term memories while also examining cross-sectionally any changes in this relationship that occur with normal ageing. This was done across two experiments. In Experiment 1, we developed a novel continuous-report task to assess short- and long-term memories for the same type of contextual-spatial associations cross-sectionally in young, middle-aged, and older adults. Experiment 2 was carried out online and aimed to replicate the findings of Experiment 1, while also minimising the potential floor effect by making minor design improvements. Both experiments were carried out with a high level of experimental control and employed STM and LTM tasks that shared a common encoding phase and were equivalent in stimulus materials and response demands. Short-term memories were probed after a delay of 8 s in the absence of any interruption, following previous research on visual short-term memories [10,20,36,37,38,42]. Intervals with durations of 15–20 min and including intervening distraction were employed to separate LTM encoding and retrieval stages [10,42,43,44].

Short- and long-term memories for contextual-spatial associations are thought to be dependent on the MTL [30,32,45,46], which is particularly vulnerable to ageing [16,17]. Here, we asked whether ageing-related decline in these associations occurred in the same way and to a similar extent within the different age groups. We hypothesised that this would indeed be the case. A shared, age-sensitive mechanism for memory decline across memory time scales would be behaviourally supported by an overall decline in both STM and LTM across the age groups, combined with similar patterns of performance deficits and a strong relationship between STM and LTM performance within each age group, and pave a path for future clinical and neuroimaging studies.

## 2. Experiment 1

### 2.1. Methods

#### 2.1.1. Participants

The study was approved by the Medical Sciences Interdivisional Research Ethics Committee of the University of Oxford in accordance with the Declaration of Helsinki (approval code: R56404/RE001). In total, 65 healthy volunteers aged 19 to 79 years took part in this study. The sample size was based on a power analysis of predicted effect sizes from a small pilot study.

Participants were members of the local community recruited via Friends of OxDARE participant registry (https://oxfordhealthbrc.nihr.ac.uk/our-work/oxdare/researchers/ accessed on 9 April 2018), the Oxford Psychology Research participant recruitment scheme, online advertisement, or word of mouth. All gave written informed consent. They either received course credit or were reimbursed for participation at a rate of £10 per hour. All participants had normal or corrected-to-normal visual acuity and normal colour vision by self-report.

#### 2.1.2. Stimuli

Stimuli were presented on a Dell OptiPlex 9030 All-in-One touch screen computer with a 1920-by-1080-pixel resolution (53.0 by 29.7 cm), using the Cogent 2000 toolbox and MATLAB release 2015b. 

Coloured photographs of complex indoor and outdoor scenes and images of everyday objects were used as stimuli. The set of scenes consisted of a combination of royalty-free images and photographs from an in-house repository purposefully selected for the experiment. It included a balanced number of indoor and outdoor scenes. Images belonged to 16 different categories (e.g., kitchen, library, forest, and beach), with 12 distinct exemplars from every category. Ninety-six scenes were randomly selected from a pool of 192 scenes for each participant. They were presented at a resolution of 1125-by-884 pixels (31 by 23.2 cm), surrounded by a grey background (RGB: [0.5, 0.5, 0.5]).

A total of 268 objects were randomly drawn from a pool of 422 coloured objects for each participant. The pool was derived from a combination of online experimental stimulus sets [47,48] and copyright-free object images available through Google Images, processed to remove backgrounds when necessary. They included a range of household, food, clothing items, toys, and sports equipment. All objects were sized to fit within a region of 75-by-75 pixels.

Objects were placed randomly within scenes with several restrictions. They were at least 50 pixels away from the edges and the centre of the screen. In three-object trials, a minimum of 500 pixels separated the centre of the objects. Objects had an equal probability of appearing in the four quadrants of the scene.

#### 2.1.3. Tasks and Procedures

The study consisted of two stages (Figure 1a). Participants reported the identity and location of objects associated with scenes based on their short-term or long-term memories of those associations.

In the first stage, participants explored scenes with one or three embedded objects for 5 s. In half of the trials, they were probed about the identity of an object in the scene and reproduced its location after a blank interval (8 s) (STM trials—Figure 1b). In the remaining half of the trials, they explored the scene in a similar fashion but were not probed for object identity and location following a blank delay (8 s) (Encoding trials—Figure 1b). These trials were employed to assess LTM for the same type of information encoded in the same way in the subsequent stage of the study.

The second stage was a surprise LTM retrieval performed after a break of approximately 20 min (LTM retrieval—Figure 1c). Participants viewed previously learned scenes and were probed about the identity of an object previously presented in the scene. They reproduced its location in a similar fashion to STM trials of the first task, but this time based on their long-term memories.

During the break, young participants completed an online questionnaire on demographic details and their general physical and mental health. During the equivalent period, middle-aged and older participants completed the ACE-III, which was used to screen participants for any signs of cognitive impairment. These two groups completed the online questionnaires at the end of the testing session.

##### Encoding and Short-Term Memory Task 

A schematic of the combined Encoding and STM trials is shown in Figure 1b. Participants were first presented with a memory array in which either one or three objects were placed within a scene. The array appeared for 5 s. Participants first had to find the embedded object(s) and remember their identity(ies) and location(s). The memory array was followed by a delay of 8 s, during which participants were presented with a blank grey screen (RGB: [0.5, 0.5, 0.5]). 

In half of the trials, the memory delay was followed by STM retrieval (STM trials). The scene from the memory array reappeared with two objects below. One of these objects had previously been embedded in the scene (target), while the other was a completely novel object from any category (foil). Participants had to identify the target object by tapping on it (identification) and then place the object as precisely as they could in its original location within the scene (localisation) using their finger by either dragging the object or tapping on the screen in the remembered location. Participants were encouraged to respond as quickly as possible when choosing the target object and to prioritise precision when reproducing its location. Once satisfied, they pressed the spacebar to proceed to the next trial. 

The other half of the trials included only the encoding of the object(s) in the scene (Encoding trials). The memory delay was followed by the presentation of the scene with a green fixation cross at the centre. In these trials, participants tapped the fixation cross and pressed the spacebar to continue to the next trial. Participants would subsequently be probed about the object-scene associations in the later, surprise long-term memory task, though they were not informed about this at this stage.

This task consisted of two blocks of 48 trials each. One-third of trials contained one object, while the remaining two-thirds contained three objects. The two trial types were randomly intermixed across the two blocks, as well as STM and Encoding trials. Therefore, at the time of encoding the identity and location of the object(s) in the scene, participants did not know when their memory would be tested. The procedure ensured that the encoding strategy and any state variables within the experimental block were equated. 

All participants completed a minimum of eight practice trials to become familiar with the procedures before the main experimental task. A separate set of scenes and objects was used for these trials. Participants could repeat the practice trials if required until they were comfortable with the task. 

##### Explicit Long-Term Memory Retrieval

Participants completed a surprise LTM task after a 20-min break (Figure 1c—LTM retrieval task). In each trial, the procedure for LTM retrieval was equivalent to that used earlier for STM trials. A previously studied scene from the Encoding trials appeared with two objects underneath. One of the objects had been previously embedded in the scene (target), while the other was a completely novel object from any category (foil). Participants first had to choose the target object (identification) and then drag it to its remembered location (localisation). Once they were satisfied with their response, they pressed the space bar. Following identification and localisation of the remembered item, participants completed two consecutive confidence rating scales, one for identification and one for localisation, respectively. They reported confidence in their performance on a continuous scale from “Not confident at all” (left) to “Very confident” (right) by moving a scale bar on the screen (results are not reported in the current paper). 

This task consisted of one block of 48 trials, which probed memories of only the Encoding trials from the previous stage. 

#### 2.1.4. Data Analysis

Data were pre-processed and analysed using MATLAB Release 2018b and R version 4.0.2 (2020) on R Studio version 1.3.1093. Trials were excluded from the analysis if identification times were faster than 100 ms or localisation occurred outside the scene boundaries (<1% of trials). Additionally, only the trials in which targets were accurately identified were included in the analysis of identification time and localisation error. Finally, participants were only included in the analysis if their STM localisation performance was within three standard deviations from the group mean. 

Data were not normally distributed. Due to the differences in skewness between STM and LTM, there were no appropriate transformations that would allow us to include data across both durations in the same parametric analyses. Therefore, non-parametric rank-based mixed factorial models were used to analyse identification accuracy, response time, and localisation error data. They were implemented using *nparLD* package version 2.1 on R. The age factor (young, middle-aged, and older groups) varied between participants. Memory duration (STM and LTM) and set size (1 and 3 objects) were manipulated within participants. ANOVA-type statistics (*ATS*) are reported due to their suitability for smaller samples, and modified *ATSs* (*ATS_Mod_*) with Box approximation were used for the between-subject factor of age group [49,50]. Where multiple comparisons were made among the three age groups, Mann–Whitney U tests were used, and the *alpha* level was Bonferroni-adjusted for the three comparisons.

Chance-level performance for localisation error was estimated by randomly shuffling each participant’s response coordinates across all trials and calculating a new localisation error from the original target location to the shuffled response location for every trial. This procedure was repeated for 1000 iterations per participant and memory type (STM, LTM). Mean localisation error was calculated across all iterations for each participant, followed by the mean across participants for each memory type, and the overall mean across memory types for the error at chance level. 

To explore the relationship between STM and LTM identification times and localisation errors across age groups, general linear models (GLMs) were fitted to mean LTM identification time and localisation error data per participant. Only data from 3-object trials were used. These trials comprised two-thirds of the overall trial number, making the resulting data more robust to noise. The initial models included mean STM identification time per participant and age group as predictor variables for LTM identification time. Mean STM localisation error per participant and age group were used as predictor variables for LTM localisation error. Both models also included an interaction term between the two predictor variables (STM identification time × Age group and STM localisation error × Age group, respectively). Based on the results, two additional simplified models were also fitted for each initial model: one excluding the interaction term and one excluding both the age group and the interaction term, with only the STM measures as predictors.

### 2.2. Results

#### 2.2.1. Participants

Five participants were excluded from the analysis: two young participants due to poor performance on one-object STM trials (outside three standard deviations from the group mean), one due to an experimenter error, and two due to Addenbrooke’s Cognitive Examination 3rd edition (ACE-III) scores < 88, which is the highest recommended cut-off to detect cognitive impairment [51]. This left a sample of 20 young (19–39 years), 20 middle-aged (41–58 years), and 20 older adults (62–79 years) (see Table 1 for demographics). The three groups were comparable in years of education. Middle-aged and older adults were also comparable in their ACE-III scores (Table 1).

#### 2.2.2. Identification Performance

There was a significant main effect of age group on identification accuracy (*ATS_Mod_*(1.83, 48.29) = 8.98, *p =* 0.001). Follow-up Mann–Whitney U tests showed that, under *alpha* = 0.017, older adults were significantly less accurate than both young (*U* = 7500.5, *z* = 3.69, *p* < 0.001) and middle-aged adults (*U* = 7208.5, *z* = 2.66, *p* = 0.008) (Figure 2a—Identification, upper panel). However, there was no difference in accuracy between young and middle-aged adults (*U* = 6711, *z* = 0.96, *p* = 0.34). 

There were also significant main effects of memory duration (*ATS*(1, ∞) = 240.25, *p <* 0.001) and set size (*ATS*(1, ∞) = 16.63, *p <* 0.001), accompanied by a significant interaction between the two factors (*ATS*(1, ∞) = 5.96, *p =* 0.01). Overall, participants were less accurate when identifying objects from LTM compared to STM and in three-item trials compared to one-item trials (Figure 2a—upper panel). Moreover, an interaction revealed the difference in accuracy between one- and three-object trials was only statistically significant in STM (*ATS*(1, ∞) = 33.76, *p <* 0.001), but not in LTM (*ATS*(1, ∞) = 0.01, *p =* 0.91). None of the other interactions reached statistical significance (see Appendix A).

A similar pattern of findings was obtained for response times in the identification report. There was a significant main effect of age group on identification time (*ATS_Mod_*(1.95, 54.24) = 11.00, *p <* 0.001). Follow-up Mann–Whitney U tests showed that, under *alpha* = 0.017, older adults were significantly slower when identifying targets than both young (*U* = 5265.00, *z* = −4.01, *p* < 0.001) and middle-aged adults (*U* = 5576.00, *z* = −2.95, *p* = 0.003) (Figure 2a—lower panel). In contrast, there was no difference in identification times between young and middle-aged adults (*U* = 6201.00, *z* = −0.82, *p* = 0.42). 

There were also significant main effects of memory duration (*ATS*(1, ∞) = 524.26, *p <* 0.001) and set size (*ATS*(1, ∞) = 102.95, *p <* 0.001), accompanied by a significant interaction between the two factors (*ATS*(1, ∞) = 50.64, *p <* 0.001). Overall, participants were slower when identifying objects from LTM than STM and in three-item trials compared to one-item trials (Figure 2a—lower panel). Moreover, the interaction revealed the difference in identification time between one- and three-object trials was only statistically significant in STM (*ATS*(1, ∞) = 141.64, *p <* 0.001) but not in LTM trials (*ATS*(1, ∞) = 1.70, *p =* 0.19). None of the other interactions reached statistical significance (see Appendix A).

#### 2.2.3. Localisation Error

There was a significant main effect of age group on localisation error (*ATS_Mod_*(1.89, 51.03) = 21.61, *p <* 0.001). Follow-up Mann–Whitney U tests showed that, under *alpha* = 0.017, and similar to identification accuracy, older adults produced significantly larger localisation errors than both young (*U* = 5434, *z* = −3.43, *p* < 0.001) and middle-aged adults (*U* = 5557, *z* = −3.01, *p* = 0.003) (Figure 2b). There was no difference in the magnitude of localisation error between young and middle-aged adults (*U* = 6303, *z* = −0.47, *p* = 0.64) (Figure 2b). 

There were also significant main effects of memory duration (*ATS*(1, ∞) = 1080.58, *p <* 0.001) and set size (*ATS*(1, ∞) = 141.98, *p <* 0.001). Moreover, the two factors of memory duration and set size interacted (*ATS*(1, ∞) = 5.85, *p =* 0.02). This interaction showed that although overall memory errors were larger in LTM and for larger set sizes, the difference between set sizes was proportionally greater in STM (*ATS*(1, ∞) = 209.26, *p <* 0.001) compared to LTM (*ATS*(1, ∞) = 40.40, *p <* 0.001). There were no other significant interactions. 

#### 2.2.4. Relationship between Short- and Long-Term Memories

To explore the relationship in performance between the two memory durations, we first ran a GLM that included STM identification response time and age group as predictors and a two-way interaction term between these two factors. The interaction did not reach statistical significance (*p* = 0.09), indicating that participants’ age did not significantly modify the relationship between STM and LTM identification response times. We. Therefore, simplified the model by excluding the interaction term. This did not significantly impair the model fit (*p* = 0.09). The resulting model, which included the STM identification response time and age group as predictors, showed that after accounting for the variation in STM identification response times, the main effect of group was not statistically significant (*p* = 0.34). The model was therefore simplified further by removing the age group. This also did not impair the model fit (*p* = 0.34). The final model demonstrated that STM identification response times alone significantly predicted LTM response times (*F*(1, 58) = 64.46, *p* < 0.001, *R*^2^ = 0.53). Participants who were slower when identifying objects from STM were also more likely to be slower when faced with LTM identification (Figure 3a).

We then ran another GLM to explore the relationship between STM and LTM localisation errors in a similar way. The initial model included STM localisation error and age group as predictors and a two-way interaction term. The interaction was not statistically significant (*p* = 0.88), indicating that participants’ age did not alter the relationship between STM and LTM localisation errors. Hence, the model was simplified by excluding the interaction term. This did not significantly impair the model fit (*p* = 0.80). The resulting model, which included the STM localisation errors and age group as predictors, showed that after accounting for the variation in STM localisation errors, the main effect of age group was no longer statistically significant (*p* = 0.94). We, therefore, simplified the model further by removing the age group. This did not impair the model fit (*p* = 0.95). The final model showed that STM localisation errors alone significantly predicted LTM localisation errors (*F*(1, 58) = 17.23, *p* < 0.001, *R*^2^ = 0.23). Namely, participants who produced larger STM errors were more likely to produce larger LTM localisation errors (Figure 3b).

In summary, both short- and long-term memories were impaired in older adults. Further, there was a relationship between memory at these two durations, which was not altered by participants’ age. 

In Experiment 2, we sought to replicate our results and go one step further in an online study by equating encoding of the short- and long-term memory trials while avoiding close-to-floor performance observed in Experiment 1 in the LTM task, especially in older adults. To this end, in Experiment 2, participants were informed about the time they would be probed on object-location associations just after encoding. More specifically, after the encoding of the memory array, a duration cue was presented. In STM trials, the “For Now” duration cue indicated that participants would be probed on their memory in that same trial. In the encoding trials, the “For Later” duration cue indicated that the participant would have to recall the information later, during the LTM retrieval stage of the experiment. Importantly, the timing of the duration cue ensured a common encoding phase for both memory durations. Lastly, to maximise the number of trials for analysis and avoid close-to-floor performance in the LTM task, we reduced the number of items to two per scene. In the interest of time, we also opted to reduce the length of the STM maintenance period to five seconds.

## 3. Experiment 2

### 3.1. Methods

#### 3.1.1. Participants

The study was approved by the Medical Sciences Interdivisional Research Ethics Committee of the University of Oxford and complied with the Declaration of Helsinki (approval code: R63062/RE001). In total, 70 volunteers across the age range participated in this study (the sample size was estimated based on pilot data). Participants were recruited via the Prolific Platform. They gave informed consent and were reimbursed at £7.5 per hour for participation. All participants had normal or corrected-to-normal vision and normal colour vision by self-report. One participant reported a Parkinson’s disease diagnosis and was excluded from the analysis. The final cohort included 69 participants (43 female) between 20 to 79 years of age, with an average of 17.4 (SD: 3.3) years in full-time education.

#### 3.1.2. Stimuli

Coloured photographs of complex indoor and outdoor scenes and images of everyday objects were used as stimuli. The scene set consisted of a combination of royalty-free images and photographs from an in-house repository. Forty-four scenes (similar to Experiment 1) were randomly selected from a pool of 399 scenes for each participant, in addition to 132 objects that were randomly drawn from a pool of 135 coloured objects. The objects (0.5 visual degrees) were selected from emoticons of everyday objects (https://emojipedia.org/ accessed on 2 June 2021) and included a range of household, food, clothing items, toys, and sports equipment.

At the start of the experimental session, participants’ screen resolution was estimated by asking them to adjust an image of a credit card to match the size of a physical credit card. We therefore could calculate the ratio between the card image width in pixels and the actual card width in centimetres. This allowed us to measure the pixel density (i.e., pixel per cm). In turn, this allowed us to control the size of the stimuli in degrees of visual angle in approximate terms by instructing participants to view the monitor at one arm’s length distance (approximately 60 cm) [52]. 

Objects were placed randomly within scenes with several restrictions. They were at least 1.5 visual degrees away from the edges and the centre of the screen. There was a minimum of 2.5 visual degrees between the centre of the objects in the two-object trials. Objects had an equal probability of appearing in the four quadrants of the scene.

#### 3.1.3. Tasks and Procedures

Similar to Experiment 1, this study consisted of two stages (Figure 4a). Participants reported the identity and location of objects associated with scenes based on their short-term or long-term memories of those associations, respectively (Figure 4).

In the first stage, participants explored scenes with two embedded objects for 5 s. After a 1-s delay, they were presented with a duration cue, indicating the time they would be probed. In half of the trials, following the “For Now” cue (0.5 s) and a blank interval (3.5 s), participants were probed about the identity of an object in the scene and reproduced its location (Figure 4b—STM trials). In the remaining half of the trials, following the “For Later” cue (0.5 s) and a blank interval (3.5 s), participants were presented with the scene without the objects embedded within them. They had to press the fixation as soon as it turned green. These trials were employed to assess LTM for the same type of information later in the study (Figure 4b—Encoding trials).

The second stage Involved an LTM-retrieval task (Figure 4c—LTM retrieval) performed after a break of approximately 15 min. Participants viewed previously learned scenes and were probed about the identity of an object previously presented in the scene. They reproduced its location in a similar fashion as in the first task (STM trials), but this time based on their long-term memories.

During the break, participants completed an unrelated task online that took around 10 min. They were encouraged to take breaks before and after this task. Importantly, the intervening task did not involve any objects or scenes. 

##### Short-Term Memory/Encoding Task 

In this stage of the experiment (Figure 4b), participants were first presented with a memory array in which two objects were placed within a scene. The array appeared for 5 s. Like Experiment 1, participants first had to find the embedded objects and remember their identities and locations. The memory array was followed by a delay of 1 s, during which participants were presented with a blank grey screen. This was then followed by a duration cue (“For Now” or “For Later” at the centre of the screen), presented for 0.5 s, indicating the duration in which participants had to keep the information in mind. 

In trials containing the “For Now” cue (Figure 4b—STM trials), participants were probed after a 3.5-s blank delay. At retrieval, the scene from the memory array reappeared with two objects below. One of these objects had previously been embedded in the scene (target), while the other was an object not seen in the scene (foil). Participants had to identify the target object by pressing the letter C to choose the object on the left or the letter M to choose the object on the right. They then had to use the mouse cursor to click the object’s original location within the scene (localisation). Participants were encouraged to respond as quickly as possible when choosing the target object and to prioritise precision when reproducing its location. Once satisfied, they pressed the mouse button to proceed to the next trial. 

In the other half of the trials, the “For Later” cue was followed by a delay (3.5 s) and the presentation of the scene (Figure 4b—Encoding trials). Participants had to click the fixation cross as soon as it turned green (after 1 s) to continue to the next trial. Participants would subsequently be probed about the object-scene associations in the later long-term memory task. 

The task consisted of two blocks of 20 trials each. The two trial types (For Now and For Later) were randomly intermixed across the two blocks. All participants completed four practice trials before the task to become familiar with the procedures. A separate set of scenes and objects was used for these trials. 

##### Explicit Long-Term Memory Retrieval 

Participants completed an LTM task after a break of approximately 15 min (Figure 4c—LTM retrieval). In each trial, the LTM retrieval stage was equivalent to that used earlier for STM retrieval. A previously studied scene from the Encoding trials appeared with two objects underneath. One of the objects had previously been embedded in the scene (target), while the other one had not been in the scene (foil). Participants had to select the target object (identification) and place it in its remembered location (localisation) using the same procedures as in the STM trials. The task consisted of one block of 20 trials. 

#### 3.1.4. Data Analysis

Data were pre-processed and analysed using MATLAB Release 2018b. Exclusion criteria were identical to those specified for Experiment 1. Repeated-measures ANOVAs with duration cue as a within-subject factor and age as a covariate were conducted for identification accuracy, identification response times, and localisation errors. Data that were not normally distributed were transformed, allowing us to use parametric statistics.

The same procedure was used to estimate chance-level performance for localisation error as in Experiment 1. To explore the relationship between STM and LTM, multiple regression models were fitted using mean LTM identification time and localisation error data, as explained previously. 

### 3.2. Results

#### 3.2.1. Identification Performance

Accuracy on the identification tasks showed that performance decreased with age overall but that the decline was relatively worse in the LTM compared to the STM task. The ANOVA revealed a significant main effect of age (*F*(1, 67) = 18.75, *p* < 0.001, η^2^_p_ = 0.22) and a significant interaction between age and memory duration (*F*(1, 67) = 4.4, *p* = 0.04, η^2^_p_ = 0.06) on identification accuracy. Memory duration did not exert a significant main effect (*p* = 0.5). Within tasks, memory decline was significant for both STM (*F*(1, 67) = 6.8 *p* = 0.011, η^2^_p_ = 0.09) and LTM (*F*(1, 67) = 13.86 *p* < 0.001, η^2^_p_ = 0.17) (Figure 5a). 

For identification response times, there were significant effects of both age (*F*(1, 67) = 28.77, *p* < 0.001, η^2^_p_ = 0.3) and memory duration (*F*(1, 67) = 14.25, *p* < 0.001, η^2^_p_ = 0.18). Responses slowed with ageing and were longer in the LTM task. There was no interaction between age and memory duration (*p* = 0.6) (Figure 5b).

Identification accuracy in the STM task was high, and its low variability precluded analysis of the relationship between performance for the two memory timescales between age groups. However, it was possible to test for the relationship using response times. We used GLM to examine whether variation in STM identification response times and age predicted LTM response times. A model including age and STM performance was a significant predictor of LTM response times, *F*(2, 66) = 19.49, *p* < 0.001, *R_2_* = 0.24. Older individuals and individuals who took longer to identify the target items in the STM task were also slow in the LTM task (Figure 6a). 

#### 3.2.2. Localisation Error

Age significantly affected the amount of error in the localisation task (*F*(1, 67) = 14.6, *p* < 0.001, η^2^_p_ = 0.18). Localisation error increased with age. There was no main effect of memory duration (*p* = 0.1) or interaction between age and memory duration (*p* = 0.08) (Figure 5c).

Like in Experiment 1, to explore whether variation in STM localisation error and age predicted long-term memory performance, we used a GLM. A model including age and STM performance was a significant predictor of LTM localisation performance, *F*(2, 66) = 8.23, *p* = 0.001 *R2* = 0.2. Both variables contributed significantly to the prediction, *p* < 0.05. Participants with a larger STM error and of increasing age were more likely to produce a larger LTM localisation error (Figure 6b). Additionally, similar to Experiment 1, STM errors significantly predicted LTM errors (*F*(1, 67) = 10.6, *p* = 0.002, *R*^2^ = 0.137). 

## 4. Discussion

The current study used novel continuous-report tasks that enabled analogous assessment of both short- and long-term memories cross-sectionally across the adult age range to investigate the presence of a relationship between the two memory systems and any changes to this relationship during healthy ageing. In our first experiment, in line with previous findings, we show that older but not middle age brings impairments in both short- and long-term memories. Importantly, there was a significant relationship between short- and long-term memory performance that remained unaltered by the participants’ age. The time it took participants to identify the target objects, and the magnitude of their localisation error in STM were strong predictors of equivalent performance measures in LTM. However, age did not change this relationship. 

In a follow-up online experiment, we replicated our main pattern of results in a version of the task that avoided near-floor performance in LTM retrieval and used a distribution of ages rather than categorical age groups. To equate encoding even further, in this experiment, participants were informed about the memory duration using a cue after items were encoded into memory and before the response. They knew, therefore, that encoded associations would remain relevant even if they were not immediately probed on these. Findings from this experiment were comparable to those from Experiment 1. Both short- and long-term memories decayed with the increasing age of participants, and the significant relationship between the two memory systems was preserved with healthy ageing. Therefore, across both experiments, we demonstrate that STM performance independently and significantly contributed to LTM performance. 

To our knowledge, this is the first time the relationship between memories at short and long durations has been assessed for the same content of spatial-contextual associations encoded in an analogous way and retrieved with equivalent task demands. The strong relationships between performance variables across the memory durations and their invariance to age observed across both experiments are consistent with considerable overlap in functional mechanisms between STM and LTM. Although we could not test directly the neural mechanisms underlying this relationship, our results align with the proposals that memories for bound representations, such as object-location associations, rely on the MTL, regardless of the memory duration [32,53,54]. The involvement of the MTL and the hippocampus in STM and LTM binding [32,53,54], as well as alterations in visual functions with age that may affect the granularity of representations [55,56,57], may explain at least some of the shared vulnerability of STM and LTM to healthy ageing. Their contributions should be quantified in future studies by using mixture modelling to estimate the proportion of mis-binding errors produced [58] and testing the perceptual ability of participants [38]. In addition, there are likely other mechanisms that contribute to performance in both memory systems, which have not been quantified in this or, in fact, other studies [10,11,42]. For example, participants’ performance may be limited by their processing speed or ability to exert selective attention in a goal-directed manner to encode, maintain, and retrieve associations. The exact degree of such contributions should also be explored in future studies.

Across both experiments, STM identification time and localisation error measures accounted for a significant proportion of variance in their LTM equivalents. This pattern of results may have originated from processing during the encoding stage, which was identical for both STM and Encoding trials, and may represent what has been termed an STM bottleneck by other authors [10,11,42], where some form of a limit on STM restricts LTM encoding. The potential involvement of the encoding or early delay stages is also somewhat supported by the significant shared variance in performance between the two memory durations irrespective of the explicit knowledge on whether (and when) memories would be tested, provided to participants during the delay period in Experiment 2 but not Experiment 1. However, the influence of intentions may be limited by the sheer number of associations to be learnt, with another study testing incidental as compared to explicit learning of object identities also finding no effect [11]. Therefore, future studies should address this question in more depth by manipulating the encoding and delay stages (e.g., by varying the encoding time or the length of the maintenance period, as in [12]) and testing the granularity of short- and long-term memories.

In addition to exploring the relationship between short- and long-term memories, our tasks allowed us to examine memory performance at different durations independently. Both the ability to recall identities and locations of objects was diminished in older age across memory durations. While in Experiment 1, middle-aged adults were comparable in performance to young adults across the board, Experiment 2 showed a more gradual decline in performance with advancing age. Findings from Experiment 1 are consistent with previous reports of STM and LTM impairments from the age of 60 from relatively small studies that treated age as a categorical variable [20,36,37,38]. In contrast, findings from Experiment 2 are comparable to those from larger cross-sectional studies, which have shown a gradual decline in STM and LTM performance across adulthood [17,22]. It is unclear exactly why such a pattern of results has emerged. It is unlikely that differences between middle-aged and older groups in Experiment 1, or across the age range in Experiment 2, are due to external factors, such as interference, as participants were assessed using identical task procedures and performed identical intervening tasks (the ACE-III and an unrelated cognitive task, respectively) during the break between the encoding and the LTM retrieval tasks. It may simply be because the age of our participants was categorical in Experiment 1 and continuous in Experiment 2. Alternatively, memory decline may indeed accelerate later in life, becoming substantial enough to be detected consistently with a relatively small cross-sectional sample after the age of 60. Notably, both middle-aged and older adults in Experiment 1 showed comparable performance on the ACE-III, a standard neuropsychological test of cognition. This provides evidence for the sensitivity and utility of continuous report tasks used in this study for measuring differences in memory functions across groups of otherwise healthy adults.

Interestingly, the memory-load manipulation, included only in Experiment 1, affected not just STM but also LTM localisation performance: participants produced larger localisation errors in three-item trials than in one-item trials. This is surprising given that, in contrast to STM, LTM is thought to have a near-unlimited capacity [47]. This finding could be explained by a limited encoding window. In the study by Brady and colleagues [47], the encoding time was proportional to the number of objects to be encoded. In contrast, Experiment 1 utilised a constant five-second window in both one- and three-item trials, during which participants searched for items within scenes and encoded their identities and locations. It is, therefore, possible that the constant encoding window of five seconds in the current study has contributed to a reduced per-item encoding in three-item as compared to one-item trials, resulting in consequences on localisation performance across both STM and LTM. In contrast, there was no difference in identification accuracy between set size conditions in LTM trials. This suggests that long-term memories for object identities were less affected by the limited encoding window than short-term memories. Potentially, other mechanisms, such as familiarity, which does not require detailed representations of stimuli [59], may have contributed to object identification performance in LTM trials. 

The already discussed lack of objective measures of visual functions and processing speed limits our understanding of mechanisms underlying age-related differences in the current study. Older adults have been shown to have difficulty ignoring distracting stimuli [60]. This, in combination with the slower processing speed [61] and the limited encoding window, may have disproportionately disadvantaged older adults while they were searching for objects in memory displays in Experiment 1. In turn, this could have led to impaired localisation performance across memory conditions, as older adults would have needed more time than the other two groups to explicitly learn object locations after the objects had been found. Future studies comparing age groups should incorporate processing speed and attention tests to complement memory tasks. However, this specific limitation was minimised in Experiment 2 by reducing the memory set size to 2 objects per scene with identical encoding times, allowing for longer processing of information at encoding per object.

Overall, we found that ageing impairs both STM and LTM performance but does not alter the significant relationship between these two memory functions. By introducing a greater level of experimental control through the utilisation of equivalent stimulus materials and response demands, we could assess memory performance for different durations in a comparable way. Importantly, our experimental design also ensured equal encoding for both STM and LTM since participants became aware of the duration only after items were encoded into memory and either at probe (Experiment 1) or during memory delay (Experiment 2). Our tasks proved to be sensitive to age-related changes in performance and enabled further examination of the relationship between STM and LTM processes, as well as any changes to this relationship in healthy ageing. However, the exact neural mechanisms and the question of whether the relationship between the two memory functions is also preserved in pathological ageing remains open; specifically in light of previous studies showing distinct patterns of performance in short- and long-term memories in individuals at risk of developing Alzheimer’s disease using distinct tests at different memory durations [44]. Furthermore, our study, as well as other studies investigating the relationship between STM and LTM [10,11,36,37,38,42,43,44], used relatively short memory durations (from seconds to minutes) to test long-term memories, often relying on intervening distractor tasks to ensure the memoranda probed were not stored in STM. It remains to be seen whether the relationship between the two memory functions changes depending on the length of time memoranda are stored in LTM and following sleep, during which memory traces may be strengthened through consolidation. These tasks provide an opportunity to examine memory and its neural underpinnings in a well-controlled manner at different timescales, from seconds to minutes, hours and even days, also enabling the study of the effects of sleep on memory, in disorders with memory deficits at their core, such as Alzheimer’s disease.

## Figures and Tables

**Figure 1 brainsci-13-00106-f001:**
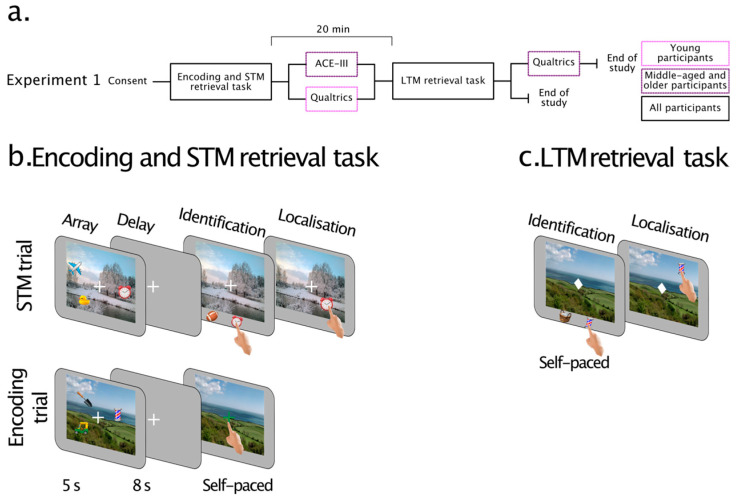
(**a**) Timeline for Experiment 1. (**b**) STM/Encoding task in Experiment 1. Each trial started with a memory array of 1 or 3 objects embedded within a scene, followed by a delay. In the short-term memory (STM) trials, the delay was followed by the STM retrieval phase. The memory scene reappeared with two objects underneath. Participants selected the object they had seen in the memory array (identification) and placed it at its original location as precisely as they could (localisation). In the encoding trials, the delay was followed by the scene containing a green fixation marker in the centre of the screen. Participants tapped on the green fixation cross. (**c**) LTM retrieval task. Participants completed a surprise LTM task after a 20 min break. In each trial, the LTM retrieval phase was equivalent to that used earlier for STM retrieval. A previously studied scene appeared with two objects underneath. Participants had to select the object previously presented in the Encoding task and drag it to its location.

**Figure 2 brainsci-13-00106-f002:**
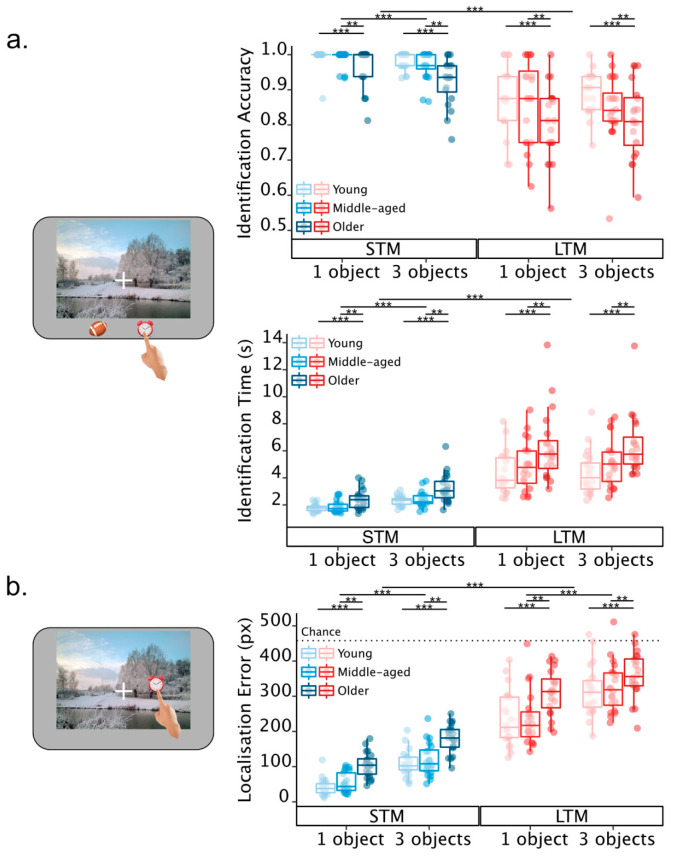
(**a**) Identification performance and (**b**) Localisation error in Experiment 1. Overall, older adults performed worse than both young and middle-aged individuals, whose performance was comparable across the tasks and for both identification and localisation. Coloured dots represent participant means, lines inside boxes represent group medians, hinges represent first and third quartiles, and whiskers represent values that fall outside the IQR but within 1.5 IQR from each hinge. *** denotes *p <* 0.001, ** denotes 0.001 < *p <* 0.01.

**Figure 3 brainsci-13-00106-f003:**
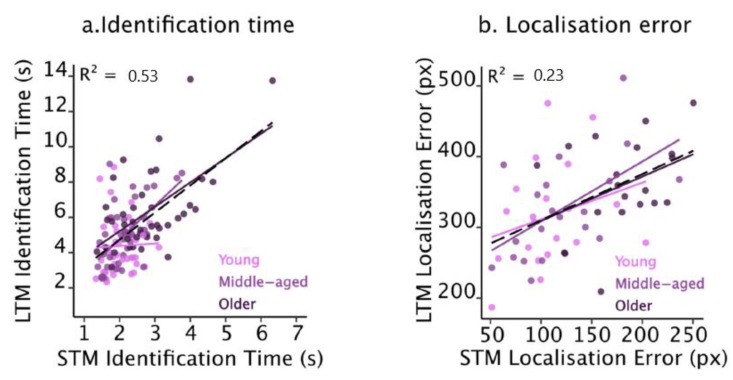
The relationship between STM and LTM for Identification times (**a**) and localisation errors (**b**) across age groups for the three object trials in Experiment 1. Coloured dots represent participant means, and coloured lines represent regression lines per age group. Dashed black lines represent model fit across groups. Regression coefficients are reported for the most parsimonious models only, which was LTM~STM in both cases.

**Figure 4 brainsci-13-00106-f004:**
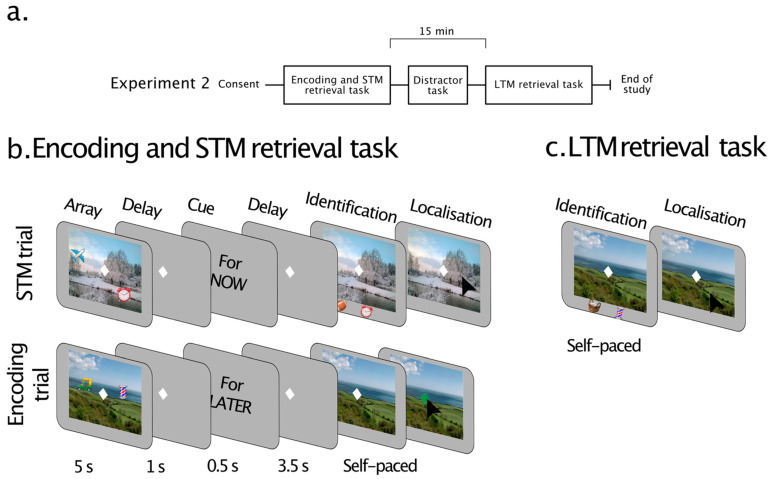
(**a**) Timeline for Experiment 2. All participants performed the same distractor task. (**b**) Encoding and STM retrieval task in 3. Each trial in the encoding and STM task started with a memory array of 2 objects embedded within a scene, followed by a delay before the presentation of the duration cue. This cue was then followed by a blank delay before the presentation of a probe. In the STM trials, the delay was followed by an STM retrieval phase. The memory scene reappeared and, after a delay, two objects appeared underneath. Participants selected the object they had seen in the memory array (identification) and indicated its original location (localisation) as precisely as possible. In the encoding trials, the delay was followed by the scene without the objects, and participants had to click the fixation as soon as it turned green. (**c**) LTM retrieval task. Participants completed an LTM task after approximately 15 min. The LTM retrieval phase was equivalent to that used earlier for the STM retrieval. A previously studied scene appeared, and after a delay, two objects were presented underneath. Participants had to identify and localise the object previously associated with the scene in the Encoding task.

**Figure 5 brainsci-13-00106-f005:**
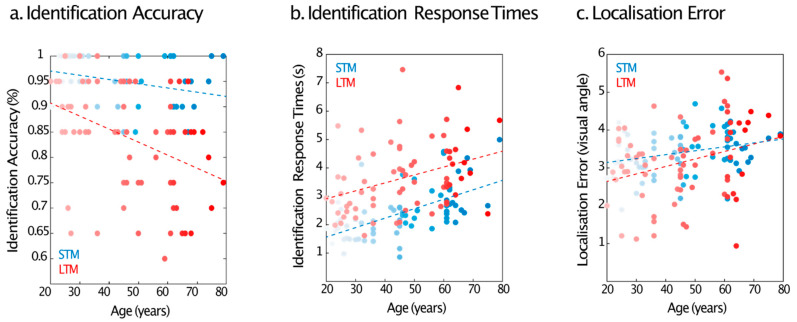
Variation of performance on both STM and LTM retrieval tasks across age in Experiment 2. Dots represent individual participants’ means (**a**) Identification Accuracy. Overall, ageing impaired performance on both memory tasks, with the worst performance in the LTM trials. (**b**) Identification Response Times. Overall, participants were slower in the LTM task and both short- and long-term memory trials were influenced by ageing. (**c**) Localisation Error. Performance was significantly less accurate for older than younger adults in identifying the location of the target object. Dots represent individual participant’s means.

**Figure 6 brainsci-13-00106-f006:**
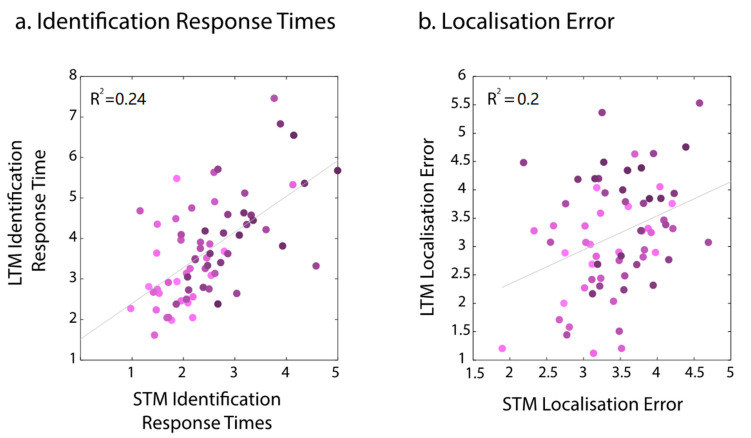
The relationship between STM and LTM for Identification times (**a**) and localisation errors (**b**) in Experiment 2. Dots represent individual participants’ means. Saturation changes reflect the age of participants.

**Table 1 brainsci-13-00106-t001:** Descriptive information. ACE-III = Addenbrooke’s Cognitive Examination III. Age, years in education and ACE-III scores are presented as mean (SD).

	Young	Middle-Aged	Older	*p*
n	20	20	20	-
Age	25.2 (4.9)	49.6 (7.0)	70.8 (4.4)	-
Male/Female	11/9	6/14	9/11	0.28
Years in education	17.8 (2.4)	17.5 (3.4)	16.8 (4.6)	0.66
ACE-III Total	-	96.5 (2.8)	96.2 (2.1)	0.67

## Data Availability

Data presented in Experiment 1 are not sharable due to ethical reasons. Data presented in Experiment 2 will be made openly available on the Open Science Framework at https://doi.org/10.17605/OSF.IO/8G5VR.

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
