# Peer review of "The Relationship between Short- and Long-Term Memory Is Preserved across the Age Range"

_brainsci, 2023, doi:10.3390/brainsci13010106_

Round 1

Reviewer 1 Report

The authors made a mistake considering as STM an 8 seconds interval and as LTM  a 20 minutes interval between training and test. 

LTM requires more time to consolidate as was decribed before. For example:

1: Kelley P, Evans MDR, Kelley J. Making Memories: Why Time Matters. Front Hum Neurosci. 2018 Oct 16;12:400. 

2: Sudhakaran IP, Ramaswami M. Long-term memory consolidation: The role of RNA-binding proteins with prion-like domains. RNA Biol. 2017 May 4;14(5):568-586.

3: Squire LR, Genzel L, Wixted JT, Morris RG. Memory consolidation. Cold Spring Harb Perspect Biol. 2015 Aug 3;7(8):a021766.

4: Jarome TJ, Helmstetter FJ. Protein degradation and protein synthesis in long-term memory formation. Front Mol Neurosci. 2014 Jun 26;7:61. 

5: Cammarota M, Bevilaqua LRM, Medina JH, Izquierdo I. Studies of Short-Term Avoidance Memory. In: Bermúdez-Rattoni F, editor. Neural Plasticity and Memory: From Genes to Brain Imaging. Boca Raton (FL): CRC Press/Taylor & Francis; 2007. Chapter 10. PMID: 21204434.

I suggest to use other times as 40-60 minutes for STM and 1 day (or more) for LTM analysis

Reviewer 2 Report

The subject of the study is very interesting, it was well designed, but it needs some adjustments to make it clearer and easier to understand, as highlighted below.

Method

I suggest the inclusion of the study approval number by the Ethics Committee

I suggest the inclusion of an experimental design timeline to facilitate the understanding of the research stages, so that all the details of the experiments are clearer regarding the chronological order of the tests.

In item 2.6 there is a mix of established parameters of inclusion and exclusion of the data and at the same time the result regarding the parameters, I suggest that this is transferred to the item results.

Result

Figure 2 - it would be interesting to include in the graph the indication of the comparisons that had results with signifcant differences, as described in the text.

I think it would be interesting to include the post hoc analysis in the supplementary material, especially of those who had significant results, between age groups.

This applies to the following analyses, being further explored

Please review the text for typos and grammatical errors.

Reviewer 3 Report

The manuscript titled “The relationship between short-and long-term memory is preserved across the age range” aims to explore a) the relationship between short-term and long-term memory and b) changes in this relationship during normal ageing. The manuscript is composed of two experiments. In both experiments young, middle-aged, and older participants were involved and were examined on their short-term and long-term memory abilities. Main results confirmed the relationship between short-term and long-term memory. Furthermore, results showed that both types of memory decrease during the ageing. Authors discussed their results in light of previous literature and gave hints for further research.

I carefully read the manuscript, and I think it may be of interest for the readers of Brain sciences. Nevertheless, I think that could be worth considering some minor points before publication. Below there are my comments and suggestions.

Introduction section

Main and specific topics relevant for the two experimental studies are correctly listed and presented. Nevertheless, it would be appropriate a) to further detail on the main hypothesis of the studies and b) provide specific aims for each experiment.

Experiment 1

Page 4, lines 160-164: Author(s) provided information about the stimuli. Were the stimuli purposely built for the experiment?

Page 4, lines 139: Two participants were excluded because of the poor performance. Why were they? Please provide an explanation for removing them from the sample.

Page 5, lines 194-196: During the break middle-aged and older participants completed the ACE-III. The ACE-III comprised memory tests. Did the Author(s) consider the interference effect?

Page 9, line 352: Which model was chosen within the GLMs?

Experiment 2

Page 13, line 512: Again, which model was chosen within the GLMs?

Discussion

The findings of the study were well discussed by the Author(s). Nevertheless, the authors could provide further discussion about the overlapping results between the two experiments.

Round 2

Reviewer 1 Report

This reviewer accord that STM and LTM definitions could be differ between molecular biology and cognitive psychology. However, the authors make a huge mistake evaluating LTM 20 minutes after acquisition as this type of memory requieres at least one hour to be consolidated. Furthermore, during the first  hour, LTMs are labile and could be easily interfered. 

The suggestion, from this reviewer, is to add a control where subjects remember (i.e. the day after training), the scences showed during the task.  

Reviewer 2 Report

The suggested changes were made by the authors and I believe that the paper is suitable for publication.

Author Response

We are very grateful to the reviewer for their positive feedback.